behaviour

competition, conflict, dominance hierarchy, interdependence, social rank

**Author for correspondence:**
Gerald G. Carter
e-mail: carter.1640@osu.edu

†These authors contributed equally to this study.

# Social dominance and cooperation in female vampire bats

Rachel J. Crisp[1,2,†], Lauren J. N. Brent[1] and Gerald G. Carter[2,3,†]

[1]Centre for Research in Animal Behaviour, College of Life and Environmental Sciences, University of Exeter, Exeter EX4 4QG, UK
[2]Smithsonian Tropical Research Institute, Balboa, Ancón, Panama
[3]Department of Evolution, Ecology and Organismal Biology, The Ohio State University, Columbus, OH 43210, USA

LJNB, 0000-0002-1202-1939; GGC, 0000-0001-6933-5501

When group-living animals develop individualized social relationships, they often regulate cooperation and conflict through a dominance hierarchy. Female common vampire bats have been an experimental system for studying cooperative relationships, yet surprisingly little is known about female conflict. Here, we recorded the outcomes of 1023 competitive interactions over food provided ad libitum in a captive colony of 33 vampire bats (24 adult females and their young). We found a weakly linear dominance hierarchy using three common metrics (Landau's $h'$ measure of linearity, triangle transitivity and directional consistency). However, patterns of female dominance were less structured than in many other group-living mammals. Female social rank was not clearly predicted by body size, age, nor reproductive status, and competitive interactions were not correlated with kinship, grooming nor food sharing. We therefore found no evidence that females groomed or shared food up a hierarchy or that differences in rank explained asymmetries in grooming or food sharing. A possible explanation for such apparently egalitarian relationships among female vampire bats is the scale of competition. Female vampire bats that are frequent roostmates might not often directly compete for food in the wild.

## 1. Introduction

Early studies of social dominance and intra-sexual competition often focused on males due in part to their greater reproductive skew and the relative ease of observing male–male competition, but female social dominance is now known to play a key role in structuring many animal societies [1,2]. Female social dominance is most obvious in cooperatively breeding species in

which dominant females can monopolize breeding by suppressing reproduction of subordinate females, and intense female competition selects for traits that are more typically male, such as large body size and bright colours [2–4]. Female social rank also shapes social structure in species where all females can breed. In many primates, where social ranks and social structures of natural populations are often well documented, reproductive success is influenced by a female's network of individualized social relationships, each involving a mix of cooperation and conflict [5–10]. Long-term field studies of rhesus macaques (*Macaca mulatta*) and baboons (*Papio* sp.) show that females form highly stable and strongly linear dominance hierarchies with clear hierarchical relationships between individuals and between matrilines [11,12], and that female social rank predicts female fitness [8,13–15]. Rank also predicts which individuals form social bonds; baboon and rhesus macaque females typically form bonds with those of a similar rank to themselves [16–21].

Many primates regulate their social relationships by social grooming, which could be related to social rank in several ways. For example, in groups with steep hierarchies, low-ranked individuals often direct grooming up the hierarchy, whereas in groups with shallow dominance hierarchies, grooming is often more symmetrical [16–29]. It remains unclear whether such patterns apply to non-primates that also form complex individualized cooperative relationships (but see [30]). The goal of this study is to assess (i) if female vampire bats have clear dominance ranks and (ii) if rank influences social grooming or food sharing.

Female common vampire bats (*Desmodus rotundus*) have been used to study the functions and development of cooperative relationships [31], but surprisingly little is known about female conflict or social rank in this species. Common vampire bats seem to share many convergent life history and social traits with primates despite their lineages diverging about 60–70 Ma [32]. When compared with other closely related bats, vampire bats have an extended longevity [33], prolonged period of offspring dependency [34–36], exceptional rates of social grooming [37–39], and high levels of social complexity, involving the formation of individualized social relationships [31,40–42].

Vampire bats live in highly fluid societies, in which compositions of roosting groups change frequently, and individuals interact to varying degrees with kin and many non-kin. In Costa Rica, colonies included several groups of 8–12 adult females and their offspring within hollow trees [43,44], but in other sites larger aggregations occur that might include cryptic subgroups [42,45]. Although female vampire bats are largely philopatric and form multiple matrilines, most females are not close kin, maternal siblings are unlikely to share fathers, and an adult female might immigrate into a group about every 2 years. As a consequence, kinship within roosts is on average low (0.07–0.08) but highly variable [42,46,47] similar to other mammals with high relationship complexity [9]. For example, the same average kinship is seen in female yellow baboons, eastern gorillas, and Asian elephants [9].

Male vampire bats have clear dominance relationships with other males that determine access to territories in roosts, such as hollows in trees or caves, that attract females [46]. A dominant male guards a territory and can father almost half the offspring of females that occupy their roosts during their tenure of up to 17 months; however, they do not appear to control individual females, which are larger than males [45,46]. Young male vampire bats disperse at 12–18 months of age, possibly in response to aggression from the resident dominant male [44,46].

Despite frequent movements between and within roosts, females tend to roost near preferred individuals that are also grooming and food-sharing partners [42–44]. Regurgitated food sharing appears to be critical for vampire bats because they regularly fail to feed in the wild and have a poor capacity to store energy [43,48]. Food donation rates are predicted by grooming, reciprocal donations and kinship [40], and new food-sharing bonds develop through escalations of reciprocal grooming [31]. Vampire bats groom each other more than other bat species and social grooming is more frequent in females than males [37,38].

Little is known about female social dominance. Agonistic interactions during feeding at wound sites suggests that dominance relationships could determine access to food. Vampire bats sometimes fight over wound sites in the wild (or feeders in captivity) and they will aggressively defend, or take over, a wound site from others [44,45,49–53]. In one small captive study, a female was dominant over two males [50], and in another captive group of six females and three male vampire bats, Park [52] observed pushing and fighting, and submissive behaviours, such as waiting for feeding bats to leave before approaching to feed. Young bats engaged in aggressive behaviours more than adults, dominant males were submissive to females and the most dominant individual fed first, but other bats did not follow a clear feeding order [52]. These observations suggest a dominance hierarchy both between and within sexes, especially in males, but no study to date has rigorously measured dominance structure or tested if social rank influences cooperative behaviours in female vampire bats.

Here, we recorded the outcomes of competitive interactions over food among a captive colony of 33 vampire bats (24 adult females and their non-reproductive offspring). We then assessed the existence and steepness of a dominance hierarchy using Landau's $h'$ measure of linearity, triangle transitivity and directional consistency. For context, we then compared these measures of female dominance with those of other species. We then tested (i) if social rank was predicted by body size, age or reproductive status, (ii) if kinship, grooming rates or food-sharing rates predicted female competitive interactions, (iii) if grooming or food sharing was directed up the hierarchy, and (iv) if asymmetries in grooming or food sharing could be explained by differences in female social rank.

# 2. Material and methods

## 2.1. Study subjects and animal care

Subjects were a captive colony of 33 common vampire bats (*Desmodus rotundus*), housed in a 2.3 × 4.5 × 2.5 m flight cage at the Smithsonian Tropical Research Institute in Gamboa, Panamá for a 2-year experiment on the development of cooperative relationships in female vampire bats [31]. The colony included 24 adult females captured in Panamá from two distant sites: Las Pavas ($n = 7$) and Tolé ($n = 17$), as well as nine young bats (four males, five females) that were born in captivity between 3 June 2016 and 15 December 2016. We sampled competitive interactions from 1 November 2016 to 31 January 2017. Bats were individually marked using a unique combination of bands of four types (coloured, round, shiny, dull) on their forearms. Bats were able to feed from a row of 3–10 spouts of blood on the floor of the cage between the hours of 18.00 and 9.00. To prevent coagulation, we added 11 g of sodium citrate and 4 g of citric acid per 4 l of bovine or porcine blood collected from a slaughterhouse.

## 2.2. Kinship and cooperative relationships

We used previously published kinship estimates that were based on known maternities and 17 microsatellite markers [42]. We also used previously published rates of dyadic grooming and food sharing from a series of fasting trials, where a focal bat was isolated and fasted alone and then introduced to the group for one hour to induce food sharing and grooming [31]. For all analyses, behaviour rates were transformed using natural log $(x + 1)$, where $x$ is the seconds of interaction per fasting trial divided by the number of trials where grooming or sharing food was possible [31]. We only used grooming and food-sharing data from the start of the colony until 31 January 2017, when we finished sampling competitive interactions, but results did not change if we instead used all observations of grooming and sharing.

## 2.3. Competitive interactions

To observe competitive interactions, we video-recorded feeders using an infrared-illuminated surveillance camera from 17.30 to 8.30 on 70 nights (1050 h). We identified 'winners' and 'losers' from five types of events at the feeders:

1. *Contact intrude*: a feeding bat is replaced at the feeder by an intruding bat using physical contact (e.g. electronic supplementary material, video S1). The intruder is the winner.
2. *No-contact intrude*: the same as 'contact intrude' but without physical contact.
3. *Contact defend*: a feeding bat uses physical contact to maintain its position at the feeder following an approach by another bat. The defender is the winner.
4. *No-contact defend*: the same as 'contact defend' but without physical contact.
5. *Waiting*: A bat in view does not begin to feed until a feeding bat leaves the feeder. The waiting bat is the loser.

From these events, we created win–lose matrices, which summed the total number of wins made by each individual against every other. We also created a win–lose matrix which combined all event types. To ensure we had a sufficient sample of observations per individual to infer ranks in a moderately steep dominance hierarchy, we followed guidelines recommended by Sánchez-Tójar et al. [54] that: (i) all bats interacted with another bat at least once, (ii) there were on average 10 sampled interaction events per bat,

a sampling effort recommended for detecting a moderately steep dominance hierarchy, and (iii) the proportion of dyads that we observed interacting (66%), was greater than what is expected from a Poisson-based null model (46%, 95% CI: 36–58%).

## 2.4. Measuring group-level dominance

For comparison with other species, we assessed dominance structure at the group level using six previously established measures, using the 'compete' R package [55]. We quantified linearity via two common metrics: *de Vries h' index* [56] and *triangle transitivity* [57]. Linearity and transitivity mean that for any given triad within the group: if A dominates B, and B dominates C, then C should not dominate A or B. Triangle transitivity can lead to more accurate estimations of linearity than Landau's *h'* when observations are sparse or when many dyads have unknown interaction types. Dominance interactions within each dyad are also expected to be asymmetric, which we quantified via *directional consistency* [58], which measures the mean frequency that a behaviour is performed in one direction, relative to the total number of times it is performed in either direction. To test for group-level dominance, we compared each of these observed metrics with the distribution (and 95% quantiles) of metrics expected under the null hypothesis, which we generated by randomizing the direction of dyadic event data 5000 times. We also fit a linear model to test how each observed metric changed with an increasing number of observations drawn without replacement in random order.

Dominance hierarchies can vary in steepness and shape. 'Despotic' hierarchies are characterized by highly linear dominance relationships and access to resources being heavily skewed towards a small proportion of dominant individuals. In 'egalitarian' hierarchies, weakly linear dominance relationships lead to lower skew in resource access [59]. These differences can be quantified by *hierarchy steepness*, which we measured as the slope of a line fitted to the relationship between individual ranks and normalized David's scores [60]. We also assessed *hierarchy shape* using a plot of how well rank differences predicted the propensity to win [54]. In addition to allowing comparisons between datasets, plotting confidence intervals on the winning probability of dominant individuals provides insight into the certainty that an estimated hierarchy reflects a real underlying dominance structure.

## 2.5. Comparisons with other species

To put our results in context, we compared group-level metrics with 14 other datasets of female social rank in mammals, and with 172 dominance interaction matrices from 84 different species ranging from invertebrates to birds to mammals, compiled by Shizuka & McDonald [57]. To assess certainty of our dominance hierarchy, we also compared measures of 'uncertainty by repeatability' and 'uncertainty by splitting' [54] to those in the 172 other datasets.

## 2.6. Assigning social rank to individuals

We assigned social ranks to individual bats using three measures: David's score, Elo-rating and Glicko-rating. *David's score* calculates individual ranks by summing wins and losses for each individual scaled to the summed scores of their interaction partners [61], a method which performs well in comparisons with others [62–64], and can be estimated from the win–lose matrix. *Elo-rating* calculates individual rank using a common numerical starting score for all individuals that is updated with each competitive interaction to give a final rank [65,66]. *Glicko-rating* [67] differs from Elo-rating in that the points which are gained or lost following an interaction are not matched in both players, but are instead adjusted according to a function of the dyad's difference in rating and their respective rating deviations.

Because Elo-rating and Glicko-rating begin by allocating all individuals with an identical starting score from which ratings diverge as interactions accumulate, there is a 'burn-in' period during which ranks are unreliable until enough observations have been recorded to reflect the true rank order, and the length of this period can be impossible to determine without prior knowledge of a hierarchy's structure [67]. Indeed, if individuals interact infrequently or the hierarchy is not steep, then the burn-in period may exceed the duration of the study [67]. Thus, without stable ranks, prior knowledge of dominance relationships or a high frequency of interactions, it may be more effective or conservative to use matrix-based ranking methods like David's score [68].

## 2.7. Predictors of individual social rank

To examine whether rank was predicted by body size, we plotted and fit general linear models to test if rank varied with either mean body mass or forearm length (standard proxies for body size in bats). Forearm length has much greater repeatability in vampire bats given that their mass changes dramatically with feeding and urination. To assess whether rank correlated with other characteristics, we also tested for rank differences by age category (adult versus young), age within young bats, maternity (mothers versus non-mothers) and source population (Las Pavas versus Tolé).

## 2.8. Comparing competitive and cooperative interactions

To test if higher-ranking bats had higher mean rates of grooming or sharing food towards other bats, we used Pearson's $r$ correlation. To test for correlations between networks, we used Mantel tests with Pearson's $r$ using the vegan R package [69]. To test two simultaneous predictors on a response, we used multiple regression quadratic assignment procedure with double semi-partialling (MRQAP) in the asnipe R package [70]. To assess whether social rank influenced the time when bats were observed at feeders (e.g. if higher-ranked females fed earlier), we used a permutation test in which we compared the observed regression coefficient for the effect of an actor's rank on event time with the expected coefficient values when each actor was re-assigned a random rank throughout the entire dataset. We used 5000 permutations in all these permutation tests.

Since bats with preferred relationships might be more likely to feed at the same time and hence compete, we used MRQAP to test if conflict rates were predicted by networks of kinship, grooming or food sharing when controlling for number of dyadic observations in the same hour. Our findings did not differ when not controlling for number of dyadic observations. To test if bats tended to groom up or down the dominance hierarchy, we made a matrix of rank difference (receiver rank–actor rank), then used MRQAP to test if dyadic grooming given was predicted by both dyadic grooming received and rank difference. By including both effects, we could test whether rank differences could help explain asymmetries in dyadic grooming. We repeated this same test with food sharing instead of grooming. Results were the same when replacing all negative rank differences with zero to test the hypothesis that bats only directed help upwards and did not discriminate between lower-ranking individuals. To test if bats tended to groom or share food with females that were closer in rank, we also conducted the same tests using the absolute rank difference.

To test if helping (grooming or food sharing) was more symmetrical in dyads of closer rank, we constructed networks of helping symmetry defined as the difference between dyadic help that was given and received divided by the total of help given and help received. We then tested the correlation between this symmetry value and absolute rank difference. For each female, we assessed the total proportion of help that was directed up the hierarchy and tested if this was different to 50%. We also measured the mean difference in the interaction rate for partners that were higher rank versus lower rank and tested if this was different than zero. In both cases, we tested this by calculating a 95% confidence interval around the mean using bootstrapping (5000 iterations, percentile method in the boot package [71,72]).

# 3. Results

## 3.1. Dominance structure at the group level

Out of 1300 recorded events at feeders, there were 1023 cases where actor and receiver bats could be identified by their bands. Metrics of group-level dominance were largely similar across the different types of events (224 contact intruding, 250 no-contact intruding, 214 contact defending, 109 no-contact defending, 219 waiting and 7 ambiguous events, electronic supplementary material and figure S1). Therefore, we combined wins from all types of events to create a single win–lose matrix (electronic supplementary material, figure S2). We observed 31 interaction events per bat (62 wins or losses, median = 46, range = 15–162).

We detected clear evidence for female–female dominance structure using de Vries $h'$ (0.30, $p < 0.0002$), directional consistency (0.59, $p < 0.0002$) and triangle transitivity (0.90, $p < 0.0002$, figure 1) when using all event types and all bats. Directional consistency and triangle transitivity were also evident when using only the data within each event type (figure 1). Results were the same when including four young males (de Vries $h'$: 0.28, $p < 0.0002$), directional consistency (0.61, $p < 0.0002$) and triangle transitivity (0.89, $p < 0.0002$).

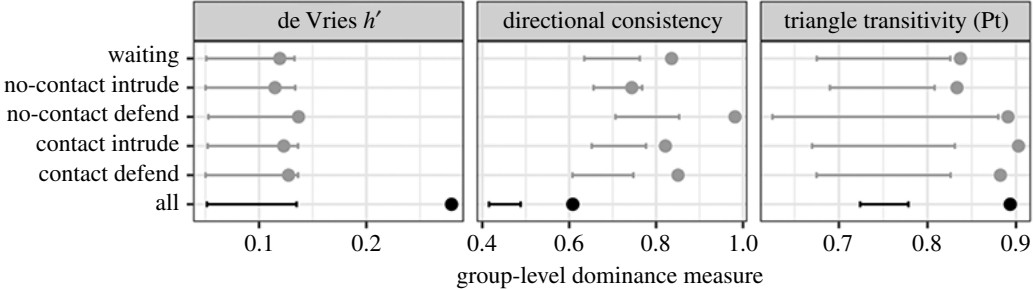

**Figure 1.** Group-level female dominance structure using three measures of dominance. Dots show observed metric and lines show the 95% confidence intervals of the metrics expected under the null hypothesis (when the direction of event data was randomized) for each event type (grey) and for all event types (black). Randomly sampling more observations led to higher de Vries $h'$ ($R^2 = 0.58$, slope = 1.9, $n = 1016$, $p < 0.0001$) and triangle transitivity ($R^2 = 0.58$, slope = 1.9, $n = 1016$, $p < 0.0001$) and lower directional consistency ($R^2 = 0.44$, slope = 0.0001, $p < 0.0001$). See electronic supplementary material, figure S1 for comparison of observed dominance measures.

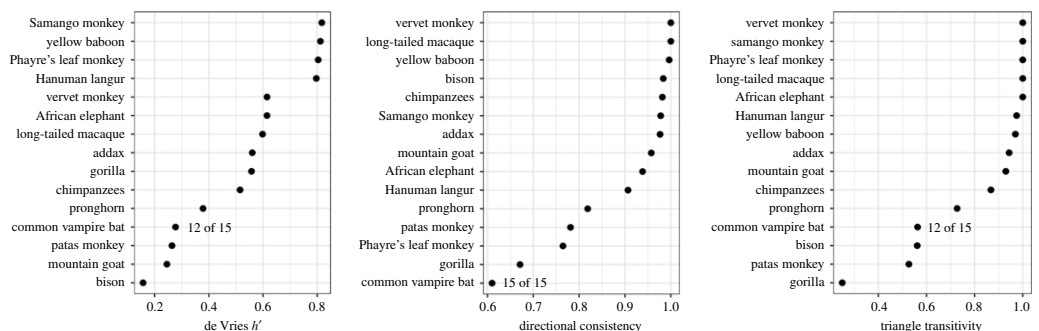

**Figure 2.** Comparison of group-level dominance measures in female vampire bats relative to 15 other datasets of female mammal groups. Data from Shizuka & McDonald [57].

Group-level dominance measures in the female vampire bats (excluding four young males) were lower than in most other groups of female mammals (figure 2). Directional consistency, triangle transitivity, and Landau's $h'$ values were in the bottom 6%, 5% and 2%, respectively, for 172 comparison taxa [57]. Likewise, uncertainty by repeatability and by splitting [54] was greater than 95% of estimates from comparison taxa, which cannot be explained by sampling effort, indicating a shallow hierarchy with relatively low confidence in individual ranks.

The female hierarchy steepness based on David's score was 0.19 ($p < 0.0001$) or 0.18 ($p < 0.0001$) when including four young males. The female hierarchy shape based on David's score showed that rank estimates did not clearly determine winning rates, especially when compared with species with a clear female dominance hierarchy (figure 3).

## 3.2. Individual ranks and predictors of rank

Individual ranks from David's score, Elo-rating and Glicko-rating were highly correlated ($n = 33$ bats including four young males, David's score versus Elo-rating: $r = 0.76$, $p < 0.001$; David's score versus Glicko-rating: $r = 0.85$, $p < 0.001$; Elo-rating versus Glicko-rating, $r = 0.93$, $p < 0.001$; electronic supplementary material, figure S4). We used David's score as ranks for subsequent analyses, because it can be replicated from the win–lose matrix, produced the clearest hierarchy shape (electronic supplementary material, figure S3), and was the most correlated overall with other methods, winning rate and losing rate. Alternate analyses with ranks based on the other two methods did not give different results, except where noted.

Young males had lower ranks on average than females ($t = -2.47$, $n = 33$, $p = 0.019$, figure 4). We detected evidence for an interaction effect between forearm size and age ($t = 3.33$, $n = 33$ bats, $p = 0.002$), because longer forearms were positively correlated with rank in the smaller young males and females ($t = 3.04$, $n = 9$, $p = 0.02$), but were negatively correlated with rank in the larger adult females ($t = -2.69$, $n = 23$, $p = 0.01$; electronic supplementary material, figure S3). The highest-ranking bats

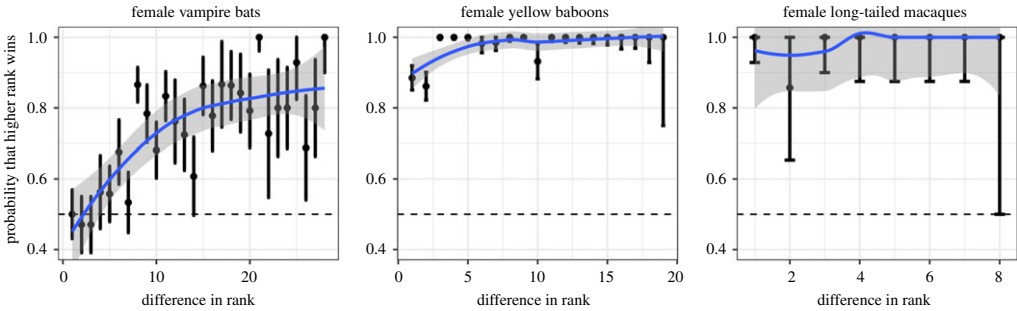

**Figure 3.** Hierarchy shape of female vampire bats compared with yellow baboons and long-tailed macaques. A greater difference in rank (David's score) should yield a higher probability that the higher rank wins. Error bars and shading are 95% CIs of points and the fit of local polynomial regression (ggplot2 R package). Dashed line shows probability expected by chance. Data from yellow baboons *Papio cynocephalus* are from Hausfater *et al.* [73]. Data from long-tailed macaques *Macaca fascicularis* are from Sterck & Steenbeek [74]. For comparison of hierarchy shape of female vampire bats by method, see electronic supplementary material, figure S3.

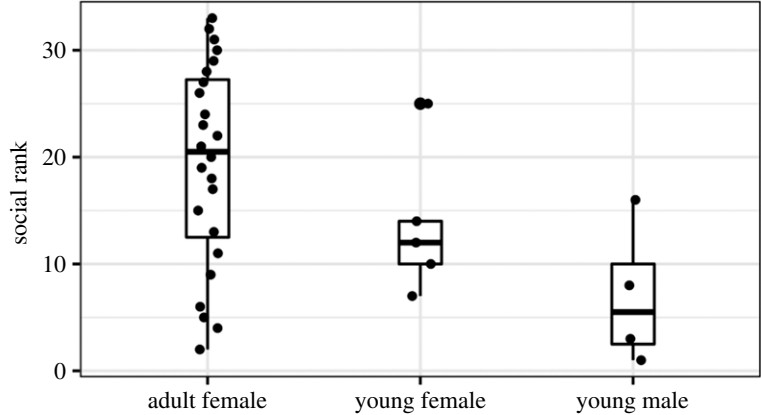

**Figure 4.** Social rank by age and sex. Rank is based on David's score (higher number is higher rank). Boxplots show the median (thick line), the first and third quartiles (box).

therefore tended to be intermediate in size (electronic supplementary material, figure S3). When excluding males, we detected only that females with smaller forearms tended to be more dominant ($t = -2.78$, $n = 29$, $p = 0.01$). Across all bats, we detected no difference in rank by body mass ($t = -1.33$, $n = 33$, $p = 0.2$), 'body condition' (mass divided by forearm, $t = 0.56$, $n = 33$, $p = 0.6$), source population ($t = -0.09$, $n = 24$, $p = 0.9$), being a mother ($t = 1.8$, $n = 33$, $p = 0.08$) or age of young bats ($t = -0.7$, $n = 9$, $p = 0.5$). We did not detect evidence that higher-ranking bats were observed at feeders earlier in the night than expected by chance ($\beta = -0.24$, $n = 2313$ observations, $p = 0.11$).

We did not find that higher-ranking adult females performed more grooming ($r = 0.15$, $n = 24$, $p = 0.5$) or food sharing ($r = -0.04$, $p = 0.9$). Among the 24 adult females, we also failed to find that competitive interaction rates were predicted by kinship ($\beta = -0.24$, $p = 0.7$), grooming ($\beta = -0.03$, $p = 0.8$), or food sharing ($\beta = -0.35$, $p = 0.2$), and results remained the same whether or not we controlled for the overlap in time at the feeders. We also did not find evidence that grooming or sharing was directed up or down the dominance hierarchy (grooming: $\beta = -0.05$, $p = 0.3$, sharing: $\beta = 0.07$, $p = 0.4$, $n = 24$). Grooming was not detectably biased towards females of closer rank ($\beta = 0.003$, $p = 0.9$). Instead, we found unexpected weak evidence that females preferentially fed females of more distant rank in either direction ($\beta = 0.18$, $p = 0.037$); but this weak trend was not robust (i.e. not detected using other rank measures or after controlling for multiple $p$-values). Finally, we did not find evidence that rank differences correlated with the symmetry of grooming ($r = -0.09$, $p = 0.9$) or sharing ($r = 0.03$, $p = 0.4$; figure 5).

## 4. Discussion

Our findings suggest that female common vampire bats form a dominance hierarchy that is weakly linear and shallow, suggesting egalitarian access to resources among familiar bats. Dominance

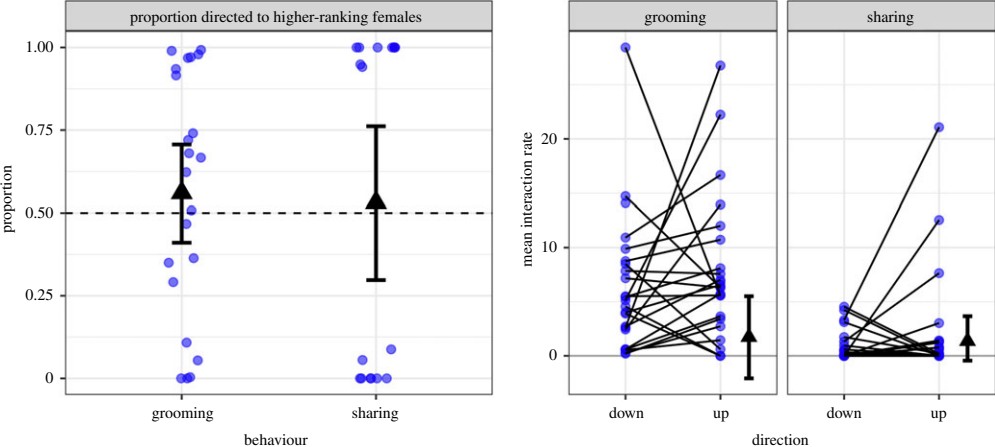

**Figure 5.** No evidence that females directed grooming or food sharing up a dominance hierarchy. Left panel shows the proportion of grooming or food sharing that each female (blue circle) directed to higher-ranking females. Triangles show means with bootstrapped 95% CI. Dashed line shows random expectation. Right panel shows the difference for each bat (line) in the mean interaction rates (circles) across female recipients that were either lower (down) or higher (up) in rank. Means with bootstrapped 95% CIs show that the overall within-bat difference is not different than zero. These plots are for visualization; actual inferences were based on more powerful permutation tests using rank differences rather than just direction.

structure among females in our study was more linear and transitive than expected from random interactions (figure 1), but also weaker relative to many other female mammals (figure 2). In this captive colony, the ranks of female vampire bats (with or without their young) were less stable, less linear, and less despotic than the majority (over 90%) of datasets from 172 other comparison species [57]. As expected with a shallow and weakly linear hierarchy, the outcomes of competitive interactions were only predictable when there was a large difference in ranks (figure 3).

Given a shallow hierarchy and inability to assign ranks with high confidence, it is not too surprising that we did not find evidence that ranks were explained using other measures. Rank was not predicted by body mass, sex, age, reproductive status or location of origin, except that young male bats had lower ranks than females (figure 4). Our findings suggest that the relationship between forearm size and rank might be nonlinear or depend on age (electronic supplementary material, figure S5), but more observations are necessary to confirm this idea. These findings are consistent with three non-mutually exclusive possibilities: female social rank might not be driven by physical traits, individual ranks might not be sufficiently precise to detect effects, and winning rates might be more influenced by other factors such as winner–loser effects [75].

Contrary to patterns in some female primates [23,24], we found no evidence that females directed grooming or food sharing to higher-ranking females, or that rates of grooming, food sharing or their symmetry, were biased to closer ranked females. We also found no clear evidence that grooming or food sharing is exchanged for tolerance from higher-ranked females (e.g. [76]), which is consistent with other evidence that grooming and food sharing promotes reciprocal investments, rather than mere tolerance [31,39,40,43]. However, a lack of a clear relationship between cooperative relationships (based on food-sharing and grooming) and rank (based on interactions at feeders) is not clear evidence for no relationship between these behaviours. The findings of our study must be interpreted with several important limitations in mind.

First, like many other studies based on visual observations, many dyads were observed interacting only a few times or were never observed to interact, so the precision of winning rates varies and many dyadic winning rates are likely to be imprecise (electronic supplementary material, figure S1). One could address this issue by forcing bats to compete over limited food in pairwise interactions.

Second, displacements and waiting at feeders might not convey dominance. If grooming is exchanged for tolerance during feeding, then this effect may negate the ability to detect dominance, because more dominant females would allow subordinates who groom them to also feed near them or even displace them from the blood spout, especially if spouts are not limited. Similarly, displacement rates could also be a poor proxy for actual dominance relationships if higher-ranking individuals make space for subordinate individuals for non-competitive reasons. For example, a dominant individual feeding alone might prefer to have a subordinate individual beside them to dilute predation risk. One could

potentially address these issues by making food more limited when observing dominance, by scoring dominance in more explicitly aggressive conflicts, or forcing bats to compete for limited spatial positions in another way (e.g. positions closer to a heat source in a cold environment).

Third, dominance interactions might be mediated in large part by vocalizations that were not recorded but do often occur at feeding sites [51]. Vampire bats can recognize other individuals at a distance by social calls [77] and possibly by echolocation calls, as reported in other bats [78–81]. Such calls might even ensure that higher-ranking and lower-ranking females rarely encounter each other at feeders, such that many clear dominance relationships might be scored as zero.

Fourth, if dominance behaviours in vampire bats are highly context-dependent, then dominance behaviours in the wild and captivity might differ in important ways. In captivity, access to food is much easier than in the natural environment, where a bat must at some risk to itself first make an open wound by biting a much larger host animal that could suddenly attack or move away, and where every wound might also be taken over by other nearby conspecifics [44,45,49,82]. By contrast, the bats in our study did not have to bite hosts, and the multiple blood-filled spouts could always be shared by individuals and were accessible during the entire night, so the benefits of competitive interactions were likely to be greatly reduced relative to the wild.

Given that these limitations would reduce evidence for dominance structures, this suggests that females do have a dominance hierarchy that is underestimated in this study. However, there are also several reasons why one might expect female vampire bats to be exceptionally egalitarian as suggested by our results. The fact that vampire bats regurgitate food to feed their close female associates suggests a lack of competition over food with preferred partners [40,43]. That is, if an individual would give food to another, then there is no reason to use rank to monopolize that food. Socially bonded female vampire bats may also have higher fitness interdependence [83], meaning that each female might benefit from the survival and reproduction of other females in her group. Fitness interdependence involves a correlation between the fitness of different individuals, but it is hard to distinguish cause from effect. Cooperative food-sharing relationships could lead to reduced competition over food among familiar bats, or alternatively, a lack of competition over food between frequent roostmates could facilitate the evolution of cooperative food-sharing relationships.

High fitness interdependence is likely if cooperation occurs 'locally' (e.g. blood sharing with familiar bats within the roost) while competition occurs 'globally' (e.g. contests over wounds with unfamiliar bats outside the roost). This scenario is consistent with observed patterns of co-roosting and foraging behaviour in the wild [43–45,49]. In species with stable groups that move together, highly associated groupmates might also be primary competitors for food, but female vampire bats might be unlikely to repeatedly compete for food with the same individuals with which they roost, groom, and share food. If female vampire bats do not experience high levels of competition within their social networks of familiar females because they compete primarily with bats from foreign groups, then studies on social foraging in vampire bats under natural conditions should reveal that frequent roostmates do not consistently hunt together (as might be expected from past observations [44]), and that most competition over wounds or prey in the wild typically involves unfamiliar vampire bats.

Testing differences in dominance rank by sex in vampire bats will require further study, but social rank in female vampire bats appears to be less clear than among males based on captive [52] and field observations [44,46]. The causes and consequences of social rank are often different for male and female mammals [2]. For instance, in some chimpanzee populations (*Pan troglodytes schweinfurthii*), males challenge others to gain rank whereas females queue [84].

Compared with other group-living mammals, social dominance is understudied in bats [85]. This lack of attention is primarily because competition over food by marked individuals is difficult to observe under natural conditions, but increasing evidence suggests bats do often compete over food [86–90], and that competitive or producer–scrounger interactions can occur repeatedly among the same individuals [91]. More comparative data on competitive interactions in bats would be necessary to test whether socioecological models that have been formulated and tested in primates [92,93] could also be applied to bats.

To conclude, we found evidence that female common vampire bats form a dominance hierarchy that appears to be weak and shallow in comparison with females in other well-studied taxa. Female social rank in captive vampire bats is less linear, steep and clear than among primates with female philopatry such as baboons and macaques. This egalitarian social structure among females is consistent with patterns of symmetrical helping and low rates of conflicts among females within the roost, and is possibly due to cooperation and competition occurring among different subsets of bats.

**Ethics.** Animal procedures were approved by the Smithsonian Tropical Research Institute Animal Care and Use Committee (#2015-0915-2018-A9 and #2017-0102-2020) and by the Panamanian Ministry of the Environment (#SE/A-76-16).

**Data accessibility.** All data and R code can be found on Figshare: https://doi.org/10.6084/m9.figshare.14043794.v1 [94].

**Authors' contributions.** R.J.C. collected the data and participated in the study design, data analysis, and writing; L.J.N.B. helped with data analysis and critically revised the manuscript; G.G.C. conceived and coordinated the study and participated in the study design, data analysis and writing. All authors gave final approval for publication and agree to be held accountable for the work performed therein.

**Competing interests.** We declare we have no competing interests.

**Funding.** R.J.C. was supported by an Ernst Mayr Short-term Fellowship from the Smithsonian Tropical Research Institute. Work by G.G.C. is supported by the National Science Foundation under grant no. 2015928.

**Acknowledgements.** We thank Rachel Page for logistical support and Tim Fawcett and James Curley for helpful discussions. We thank the editor Oliver Schülke and two anonymous reviewers for feedback that improved the manuscript.

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
