## [Peer Review File · Royal Society Open Science]

Review History

RSOS-210266.R0 (Original submission)

Review form: Reviewer 1

Is the manuscript scientifically sound in its present form?

Yes

Are the interpretations and conclusions justified by the results?

Yes

Is the language acceptable?

Yes

Do you have any ethical concerns with this paper?

No

Have you any concerns about statistical analyses in this paper?

No

Recommendation?

Accept with minor revision (please list in comments)

Comments to the Author(s)

This study examines dominance among female vampire bats, who show convergence with primates in many social and life-history traits. The authors monitored agonistic interactions over access to an ad lib food supply. Although there was evidence of a linear and transitive dominance hierarchy, dominance appears to be attenuated compared to many other species. The authors found that the adult female dominance hierarchy was shallow, that rank was not associated with any of their tested covariates, and that rank relationships did not appear to structure cooperative relationships.

I very much enjoyed this thorough and insightful study. I have minor suggestions for improvement.

l.20 A primary caveat of the study is that dominance was measured in the context of competition over an ad lib food supply, which could lead to reduced competition and thus reduced impact of dominance. As such, I think the abundance of food should be noted in the abstract somewhere. Perhaps line 20 could read 'competitive interactions over food (provided ad lib)' or something similar.

l. 23-24 "than many" should be "than in many"

l. 89 – I recommend the authors provide some interpretation of these values so it is clear what point the authors are trying to make. Is this higher/lower/comparable to average relatedness in groups of the primate species discussed in prior paragraphs?

l. 92 – Is "recruit" in this sentence used to refer to immigration, reproduction, or immigration + reproduction?

l. 95-97 – Is it known whether dominance in male vampire bats is associated with the potential predictors of dominance in females tested in this study (age, body size)? If so, it would be useful to mention here what is known about the basis of dominance in males.

l. 127 "to that of" should be "to those of"

l.206 – 207: Perhaps the authors can elaborate on what they mean here. How does measuring steepness and shape of the hierarchy shed light on uncertainty regarding the estimated hierarchy?

l. 212 – "Macdonald" should be "McDonald"

l.222 – The authors should provide some more detail on how these methods were implemented. There are variants to these methods and relevant parameters (Dij vs Pij for David's Score, K parameter for Elo, starting scores, etc.) that should be reported here or in the supplement.

l.224-226 – the authors introduce Glicko rating because of its ability to assess the confidence of rank estimates, but then they never use this confidence estimate in the remainder of the paper. Could these estimates of uncertainty add some additional insight to your results?

l.256-257 – The authors should provide more detail about this custom permutation. At first I thought the subsequent sentence was doing this, but after re-reading these sentences I understood that they actually relate to different questions (1st sentence: do higher-ranked females feed earlier; 2nd sentence: are conflict rates predicted by kinship or other networks). So additionally, it would be good to make it clearer that the second sentence is describing a subsequent test, not explaining the test introduced in the previous sentence.

l. 265-266 – this sentence is confusing because the authors' justification for this analysis (explain asymmetries) is immediately contradicted by results presented in the following clause (very little asymmetry). My suggestion is to remove the result, which confuses the sentence and shouldn't really be in the Methods section anyway.

l. 275-278 – This explanation of helping up the hierarchy is confusing. Could the authors reword this or break it into multiple sentences?

l. Fig 2 – I was curious about the criteria for the selection of these 15 species, so I had a look at some of these studies. In my investigation, I found that the Jenks et al (1995) matrix contains males, so I believe it should not be included in this analysis.

l. 324 – "than the" should be "that the"

l.336-337 – I'd add "except where noted" because the authors do note somewhere a difference in results based on ranking methods.

l. 358 – "and or" should be "or"

Figure 4. Why does the y-axis range between -100 and 100? The authors use normalized David's score, which should range from 0 to N, where N is the number of individuals in the group. Or are these un-normalized scores?

Figure 5. The y-axis and figure caption suggest that the left panel indicates that some individuals spend 100% of their time sharing food up the hierarchy. Is it not instead that some individuals direct 100% of their food sharing time up the hierarchy, while others direct none of their food sharing time up the hierarchy?

l. 412 – "this making" should be "this by making"

l. 440-442 – A related point that I would like to see discussed (I think it could fit here or in prior paragraphs) is that vampire bats share the exact food that they are competing over. It would make sense to me that a dominant would step aside to allow a subordinate individual to feed if the alternative was that that dominant had to regurgitate blood to the same subordinate a few hours later. I guess what I'm saying is that if the "currency" of cooperation and competition is identical, then there is less reason for dominants to use their preferred social status to monopolize the currency. A way to resolve this would be to test dominance in contexts other than competition over food (for instance, competition over heat source as suggested on line 414).

Discussion – One point I found missing from the Discussion was what implications this attenuated hierarchy has for the biology of vampire bats or dominance in general. If these results are representative of vampire bat behavior, what does this mean about the nature of competition in this species? The discussion on lines 432-452 kind of gets at this point, but it is mostly framed in the context of why the results of the study might be expected. Maybe this paragraph can be split to have a dedicated paragraph interpreting the significance of the results of the paper.

Review form: Reviewer 2

Is the manuscript scientifically sound in its present form?

Yes

Are the interpretations and conclusions justified by the results?

Yes

Is the language acceptable?

Yes

Do you have any ethical concerns with this paper?

No

Have you any concerns about statistical analyses in this paper?

Yes

Recommendation?

Accept as is

Comments to the Author(s)

I enjoyed reading the manuscript. I have no comments. The doubts that I was writing down are evacuated later in the discussion in the part that the authors warn about the considerations of the study.

My impression with vampires is that spatial memory to find food must be a valuable resource within a group. This spatial memory could be better with experience (age?) And this could possibly lead to hierarchies, however it is difficult to prove this in captive designs.

Decision letter (RSOS-210266.R0)

Dear Ms Crisp

The Editors assigned to your paper RSOS-210266 "Social dominance and cooperation in female vampire bats" have now received comments from reviewers and would like you to revise the paper in accordance with the reviewer comments and any comments from the Editors. Please note this decision does not guarantee eventual acceptance.

Please submit your revised manuscript and required files (see below) no later than 21 days from today's (ie 07-Apr-2021) date. Note: the ScholarOne system will 'lock' if submission of the revision is attempted 21 or more days after the deadline. If you do not think you will be able to meet this deadline please contact the editorial office immediately.

on behalf of Dr Oliver Schülke (Associate Editor) and Kevin Padian (Subject Editor)
openscience@royalsociety.org

Editor comments:

Thanks for your submission. As you see the two reviewers were overall happy with the manuscript but the AE had some methodological problems with it and does not think that the data can test the hypothesis. I urge you to respond to each of his and the reviewers' points, and if you need additional time, please notify the Editorial Office. Best wishes with your revisions.

Associate Editor Comments to Author (Dr Oliver Schülke):

Associate Editor: 1

Comments to the Author:

Dear Dr. Crisp, Lauren and Gerry,

please find below comments on your manuscript from two expert reviewers. As the associate editor handling your submission I am obliged to take a good look at the paper as well. I agree with the reviewers that there are a lot of good things to say: the writing is clear, the broader question (female dominance relationships and effects on affiliation and cooperation) is well derived from the literature, the reader has all the info needed on the species studied, the methods have just the right level of detail and the analyses are sound, the presentation is clear. I do have one major issue with the piece though concerning how the analyses are framed in theory as a test of Seyfarth's grooming model and/or vs? Biological Market Theory (naming them in the intro helps discussing them). It remains unclear whether the two models are pitched against each other as alternatives or not. The predictions listed on line 127ff (giving benefits up the hierarchy and negative correlation between rank difference and balance of gives and takes) can be derived from both models. It seems important to note that changes in grooming pattern with changing dominance hierarchy steepness are also not predicted from BM theory only; the ability to provide goods and services (he mainly thought of agonistic support) that others cannot provide lies at the heart of Seyfarth's model. Barrett and Henzi have also pointed out that any test of Seyfarth's predictions should be accompanied by a test of his assumptions – that grooming received in excess to given is correlated with support provided (or any other good or service provided).

But even with additional analyses, I fear that these models cannot be tested with the data the study produced, because dominance measures are conflated with the distribution of goods and services. If access to food can be monopolized but also shared and dominance is exclusively assessed via priority of access to monopolizable food, resulting hierarchy characteristics will be different from those of a formal dominance hierarchy derived from dominance interactions without any obvious utility. Work on BM has shown that grooming is often interchanged for tolerance in feeding contexts. If the same trade is at work in the vampire bats, dominants will grant those subordinates access to food who provide(d) them with more grooming. If dominance is then assessed from these interactions, the dominant of the two bats would look less dominant, because the subordinate does not have to wait, is not aggressed, can join in or even take over the blood spout of the dominant bat. I have a hard time predicting what the emerging dominance hierarchy would look like, if the inter-change of grooming for tolerance was one important feature of bat social trades.

I will be happy to reconsider a new version of your manuscript with a new theoretical angle. In case you chose to motivate the study as an investigation into female dominance structure (again), please consider analyzing only f-f interactions.

Best wishes,
Oliver Schülke

Reviewer comments to Author:

Reviewer: 1

Comments to the Author(s)

This study examines dominance among female vampire bats, who show convergence with primates in many social and life-history traits. The authors monitored agonistic interactions over access to an ad lib food supply. Although there was evidence of a linear and transitive dominance hierarchy, dominance appears to be attenuated compared to many other species. The authors found that the adult female dominance hierarchy was shallow, that rank was not associated with any of their tested covariates, and that rank relationships did not appear to structure cooperative relationships.

I very much enjoyed this thorough and insightful study. I have minor suggestions for improvement.

l.20 A primary caveat of the study is that dominance was measured in the context of competition over an ad lib food supply, which could lead to reduced competition and thus reduced impact of dominance. As such, I think the abundance of food should be noted in the abstract somewhere. Perhaps line 20 could read 'competitive interactions over food (provided ad lib)' or something similar.

l. 23-24 "than many" should be "than in many"

l. 89 – I recommend the authors provide some interpretation of these values so it is clear what point the authors are trying to make. Is this higher/lower/comparable to average relatedness in groups of the primate species discussed in prior paragraphs?

l. 92 – Is "recruit" in this sentence used to refer to immigration, reproduction, or immigration + reproduction?

l. 95-97 – Is it known whether dominance in male vampire bats is associated with the potential predictors of dominance in females tested in this study (age, body size)? If so, it would be useful to mention here what is known about the basis of dominance in males.

l. 127 "to that of" should be "to those of"

l.206 – 207: Perhaps the authors can elaborate on what they mean here. How does measuring steepness and shape of the hierarchy shed light on uncertainty regarding the estimated hierarchy?

l. 212 – "Macdonald" should be "McDonald"

l.222 – The authors should provide some more detail on how these methods were implemented. There are variants to these methods and relevant parameters (Dij vs Pij for David's Score, K parameter for Elo, starting scores, etc.) that should be reported here or in the supplement.

l.224-226 – the authors introduce Glicko rating because of its ability to assess the confidence of rank estimates, but then they never use this confidence estimate in the remainder of the paper. Could these estimates of uncertainty add some additional insight to your results?

l.256-257 – The authors should provide more detail about this custom permutation. At first I thought the subsequent sentence was doing this, but after re-reading these sentences I understood that they actually relate to different questions (1st sentence: do higher-ranked females feed earlier; 2nd sentence: are conflict rates predicted by kinship or other networks). So additionally, it would be good to make it clearer that the second sentence is describing a subsequent test, not explaining the test introduced in the previous sentence.

l. 265-266 – this sentence is confusing because the authors' justification for this analysis (explain asymmetries) is immediately contradicted by results presented in the following clause (very little asymmetry). My suggestion is to remove the result, which confuses the sentence and shouldn't really be in the Methods section anyway.

l. 275-278 – This explanation of helping up the hierarchy is confusing. Could the authors reword this or break it into multiple sentences?

l. Fig 2 – I was curious about the criteria for the selection of these 15 species, so I had a look at some of these studies. In my investigation, I found that the Jenks et al (1995) matrix contains males, so I believe it should not be included in this analysis.

l. 324 - "than the" should be "that the"

l.336-337 - I'd add "except where noted" because the authors do note somewhere a difference in results based on ranking methods.

l. 358 - "and or" should be "or"

Figure 4. Why does the y-axis range between -100 and 100? The authors use normalized David's score, which should range from 0 to N, where N is the number of individuals in the group. Or are these un-normalized scores?

Figure 5. The y-axis and figure caption suggest that the left panel indicates that some individuals spend 100% of their time sharing food up the hierarchy. Is it not instead that some individuals direct 100% of their food sharing time up the hierarchy, while others direct none of their food sharing time up the hierarchy?

l. 412 - "this making" should be "this by making"

l. 440-442 - A related point that I would like to see discussed (I think it could fit here or in prior paragraphs) is that vampire bats share the exact food that they are competing over. It would make sense to me that a dominant would step aside to allow a subordinate individual to feed if the alternative was that that dominant had to regurgitate blood to the same subordinate a few hours later. I guess what I'm saying is that if the "currency" of cooperation and competition is identical, then there is less reason for dominants to use their preferred social status to monopolize the currency. A way to resolve this would be to test dominance in contexts other than competition over food (for instance, competition over heat source as suggested on line 414).

Discussion - One point I found missing from the Discussion was what implications this attenuated hierarchy has for the biology of vampire bats or dominance in general. If these results are representative of vampire bat behavior, what does this mean about the nature of competition in this species? The discussion on lines 432-452 kind of gets at this point, but it is mostly framed in the context of why the results of the study might be expected. Maybe this paragraph can be split to have a dedicated paragraph interpreting the significance of the results of the paper.

Reviewer: 2

Comments to the Author(s)

I enjoyed reading the manuscript. I have no comments. The doubts that I was writing down are evacuated later in the discussion in the part that the authors warn about the considerations of the study.

My impression with vampires is that spatial memory to find food must be a valuable resource within a group. This spatial memory could be better with experience (age?) And this could possibly lead to hierarchies, however it is difficult to prove this in captive designs.

===PREPARING YOUR MANUSCRIPT===

===PREPARING YOUR REVISION IN SCHOLARONE===

- Ensure that your data access statement meets the requirements at <https://royalsociety.org/journals/authors/author-guidelines/#data>. You should ensure that you cite the dataset in your reference list. If you have deposited data etc in the Dryad repository, please include both the 'For publication' link and 'For review' link at this stage.
- If you are requesting an article processing charge waiver, you must select the relevant waiver option (if requesting a discretionary waiver, the form should have been uploaded at Step 3 'File upload' above).
- If you have uploaded ESM files, please ensure you follow the guidance at <https://royalsociety.org/journals/authors/author-guidelines/#supplementary-material> to include a suitable title and informative caption. An example of appropriate titling and captioning may be found at https://figshare.com/articles/Table_S2_from_Is_there_a_trade-off_between_peak_performance_and_performance_breadth_across_temperatures_for_aerobic_scope_in_teleost_fishes_/3843624.

Author's Response to Decision Letter for (RSOS-210266.R0)

See Appendix A.

Decision letter (RSOS-210266.R1)

Dear Ms Crisp

On behalf of the Editors, we are pleased to inform you that your Manuscript RSOS-210266.R1 "Social dominance and cooperation in female vampire bats" has been accepted for publication in Royal Society Open Science subject to minor revision in accordance with the referees' reports. Please find the referees' comments along with any feedback from the Editors below my signature.

Please submit your revised manuscript and required files (see below) no later than 7 days from today's (ie 04-May-2021) date. Note: the ScholarOne system will 'lock' if submission of the revision is attempted 7 or more days after the deadline. If you do not think you will be able to meet this deadline please contact the editorial office immediately.

Please note article processing charges apply to papers accepted for publication in Royal Society Open Science (<https://royalsocietypublishing.org/rsos/charges>). Charges will also apply to papers transferred to the journal from other Royal Society Publishing journals, as well as papers

submitted as part of our collaboration with the Royal Society of Chemistry (<https://royalsocietypublishing.org/rsos/chemistry>). Fee waivers are available but must be requested when you submit your revision (<https://royalsocietypublishing.org/rsos/waivers>).

on behalf of Dr Oliver Schülke (Associate Editor) and Kevin Padian (Subject Editor)
openscience@royalsociety.org

Associate Editor Comments to Author (Dr Oliver Schülke):

Comments to the Author:

Dear Gerry, Lauren and Dr. Crisp,

thanks for making the effort to revise the paper for RSOS. I am impressed by your turn-around times!

I still have some comments regarding the framing of the study. If you are able to resolve these issues I will be happy to accept it.

Line 45: Instead of just saying "For example" here I suggest to introduce primates as an order where this has been extensively studied. Then it is clear why you stick with primates for a bit.

Line 66ff (line nrs. refer to the version with racked changes): Here you simply state how grooming patterns change with steepness of the dominance hierarchy but you do not say why this would be the case. You say with low hierarchies you benefit more from grooming who grooms you most. What is the benefit of that and what is the benefit of grooming up the hierarchy in a steeper system? You talk about rank regulating access to resources but there is one logical step missing which is tolerance or passive food sharing being interchanged for grooming. If you are to discuss the lack of direct competition over food as the reason for the lack of a clearly developed dominance hierarchy in your bats, you need to explain here what competition has to do with all of this. But since you do not want to go there (see below), you better omit any mentioning of benefits or competition here – just describe the patterns.

Line 71: I understand now what you are trying to do, but I still have problems with that. I think that you still do not say clearly enough what you want to do. You still say: It remains unclear whether these primate-based models can be applied to non-primates. If you are to answer this, you need to say what "these models" are. If you want to understand whether they are applicable, you do not need to test their predictions, you need to assess their assumptions and inner workings and decide whether vampire bat biology fits those (see also line 468 in discussion). If you decide they do, the models can be applied and the next step is to test their predictions. BUT I think what you are really doing is testing whether similar patterns of grooming distribution according to dominance rank can be found in bats. Which would be fine, but not if you want to discuss them in relation to competition. If you want to go there, you have to explain the role of competition in "these models".

Line 73: If these are indeed your goals, you do not need any theory or models for that. You can simply say that hierarchy characteristics have been extensively studied in other species, that dominance rank can have profound effects on the exchange of grooming and other goods and services which may be modulated by how steep the hierarchy is. This way you stay on the phenomenological level, you can maintain your aims and you do not have any troubles with how you present the models and how they relate to this work (being assessed for applicability, being

tested, or else). This would also go well with the tests you on lines 158ff. You could also largely maintain your discussion. Just describe the patterns found and how they relate to what is known from other species and then say that competition for resources and services is a key factor in theoretical models and that you have not studied this here but that you want to explore in the following where this could go.

I am sorry for creating extra work for you but I really think that a bit more clarity is needed here to guide your readers and make them take away a maximum of info.

Take care,

Ole

===PREPARING YOUR MANUSCRIPT===

===PREPARING YOUR REVISION IN SCHOLARONE===

Please ensure that you include a summary of your paper at Step 2 'Type, Title, & Abstract'. This should be no more than 100 words to explain to a non-scientific audience the key findings of your

research. This will be included in a weekly highlights email circulated by the Royal Society press office to national UK, international, and scientific news outlets to promote your work.

Author's Response to Decision Letter for (RSOS-210266.R1)

See Appendix B.

Decision letter (RSOS-210266.R2)

Dear Ms Crisp,

I am pleased to inform you that your manuscript entitled "Social dominance and cooperation in female vampire bats" is now accepted for publication in Royal Society Open Science.

on behalf of Dr Oliver Schülke (Associate Editor) and Kevin Padian (Subject Editor)
openscience@royalsociety.org

Follow Royal Society Publishing on Twitter: @RSocPublishing
Follow Royal Society Publishing on Facebook:
<https://www.facebook.com/RoyalSocietyPublishing.FanPage/>

Read Royal Society Publishing's blog:
<https://royalsociety.org/blog/blogsearchpage/?category=Publishing>

Appendix A

Response to reviewer comments

Editor comments:

Thanks for your submission. As you see the two reviewers were overall happy with the manuscript but the AE had some methodological problems with it and does not think that the data can test the hypothesis. I urge you to respond to each of his and the reviewers' points, and if you need additional time, please notify the Editorial Office. Best wishes with your revisions.

Our responses are in bold. We thank the editor and reviews for their highly constructive feedback. We feel we have addressed all concerns about the original manuscript.
-Gerry Carter, Laurent Brent, and Rachel Crisp

Associate Editor Comments to Author (Dr Oliver Schülke):

Associate Editor: 1

Comments to the Author:

Dear Dr. Crisp, Lauren and Gerry,

please find below comments on your manuscript from two expert reviewers. As the associate editor handling your submission I am obliged to take a good look at the paper as well. I agree with the reviewers that there are a lot of good things to say: the writing is clear, the broader question (female dominance relationships and effects on affiliation and cooperation) is well derived from the literature, the reader has all the info needed on the species studied, the methods have just the right level of detail and the analyses are sound, the presentation is clear. I do have one major issue with the piece though concerning how the analyses are framed in theory as a test of Seyfarth's grooming model and/or vs? Biological Market Theory (naming them in the intro helps discussing them). It remains unclear whether the two models are pitched against each other as alternatives or not. The predictions listed on line 127ff (giving benefits up the hierarchy and negative correlation between rank difference and balance of gives and takes) can be derived from both models. It seems important to note that changes in grooming pattern with changing dominance hierarchy steepness are also not predicted from BM theory only; the ability to provide goods and services (he mainly thought of agonistic support) that others cannot provide lies at the heart of Seyfarth's model. Barrett and Henzi have also pointed out that any test of Seyfarth's predictions should be accompanied by a test of his assumptions – that grooming received in excess to given is correlated with support provided (or any other good or service provided).

Reading these comments, we see that we may have failed to properly convey or frame the questions in our introduction. Our aim was never to test these two models of primate grooming or pit them against each other. In fact, we were more interested in general and uncontroversial predictions from both models that could be applied to other non-primate species (vampire bats). We mainly reviewed this material to provide a backdrop to the existing theory on the topic of social rank and grooming/cooperation. In many primates, social rank affects grooming, but this has not been seen much or at all outside of primates. As we say in the original introduction, "it remains largely unclear how well these hypotheses can be applied to non-primates that form complex individualized social relationships".

Our aim here was to test (1) whether female vampire bats have a social dominance hierarchy, (2) how strong, clear, and linear the hierarchy is compared to other species, and (3) whether social rank is a clear predictor of grooming or food sharing in vampire bats. We answered all of these questions, and that is the contribution we make to the field in this study.

To clarify the theoretical angle of the paper, this revised section of the introduction now reads:

Many primates regulate their social relationships by social grooming, which could be related to social rank in several ways. For example, the relationship between social rank and social grooming might depend on the “steepness” of the dominance hierarchy (Barrett et al. 1999; Barrett et al. 2002). In steep hierarchies where social rank determines access to resources like food, low-ranked individuals might benefit from directing grooming up the hierarchy (Seyfarth 1977; Schino 2001; Schino 2007), but in shallow dominance hierarchies where resources cannot be monopolised, they might benefit more from grooming those who groom them more (Barrett et al. 2002; de Vries et al. 2005; Schino and Aureli 2008; Jaeggi et al. 2010: 201; Kaburu and Newton-Fisher 2015). It remains unclear whether these primate-based models can be applied to non-primates that also form complex individualized cooperative relationships (but see (Picard et al. 2020)). The goal of this study is to assess (1) if female vampire bats have clear dominance ranks and (2) if rank influences social grooming or food sharing.

We also added the following to the discussion:

However, a lack of a clear relationship between cooperative relationships (based on food-sharing and grooming) and rank (based on displacements at feeders) is not clear evidence for no relationship between these behaviours or that these primate-based models cannot be applied to vampire bats. The findings of our study must be interpreted with several important limitations in mind.

But even with additional analyses, I fear that these models cannot be tested with the data the study produced, because dominance measures are conflated with the distribution of goods and services. If access to food can be monopolized but also shared and dominance is exclusively assessed via priority of access to monopolizable food, resulting hierarchy characteristics will be different from those of a formal dominance hierarchy derived from dominance interactions without any obvious utility. Work on BM has shown that grooming is often interchanged for tolerance in feeding contexts. If the same trade is at work in the vampire bats, dominants will grant those subordinates access to food who provide(d) them with more grooming. If dominance is then assessed from these interactions, the dominant of the two bats would look less dominant, because the subordinate does not have to wait, is not aggressed, can join in or even take over the blood spout of the dominant bat. I have a hard time predicting what the emerging dominance hierarchy would look like, if the inter-change of grooming for tolerance was one important feature of bat social trades.

We agree that this is a possibility, and we now address in the discussion section on limitations:

Second, displacements and waiting at feeders might not convey dominance. If grooming is exchanged for tolerance during feeding, then this effect may negate the ability to detect dominance, because more dominant females would allow subordinates who groom them to also feed near them or even displace them from the blood spout, especially if spouts are not limited. Similarly, displacement rates could also be a poor proxy for actual dominance relationships if higher-ranking individuals made space for subordinate individuals for non-competitive reasons. For example, a dominant individual feeding alone might prefer to have a subordinate individual besides them to dilute

predation risk. One could potentially address these issues by making food more limited when observing dominance, by scoring dominance in more explicitly aggressive conflicts, or forcing bats to compete for limited spatial positions in another way (e.g. positions closer to a heat source in a cold environment).

I will be happy to reconsider a new version of your manuscript with a new theoretical angle. In case you chose to motivate the study as an investigation into female dominance structure (again), please consider analyzing only f-f interactions.

We now present all the main analyses with only female-female interactions. We also repeated some analyses also including four young males for two reasons. First, young males prior to dispersal are a central part of the maternity colony social network given their strong rates of association, grooming, and food sharing with their mothers and sometimes other females. Second, it is informative that young males were lower in rank than adult and young females (Figure 4). To clarify this, we now note whether the young males are included or excluded for every analysis in the results.

Best wishes,
Oliver Schülke

Reviewer comments to Author:
Reviewer: 1

Comments to the Author(s)

This study examines dominance among female vampire bats, who show convergence with primates in many social and life-history traits. The authors monitored agonistic interactions over access to an ad lib food supply. Although there was evidence of a linear and transitive dominance hierarchy, dominance appears to be attenuated compared to many other species. The authors found that the adult female dominance hierarchy was shallow, that rank was not associated with any of their tested covariates, and that rank relationships did not appear to structure cooperative relationships.

I very much enjoyed this thorough and insightful study. I have minor suggestions for improvement.

We thank the reviewer for their kind words and highly constructive suggestions for improvement.

I.20 A primary caveat of the study is that dominance was measured in the context of competition over an ad lib food supply, which could lead to reduced competition and thus reduced impact of dominance. As such, I think the abundance of food should be noted in the abstract somewhere. Perhaps line 20 could read 'competitive interactions over food (provided ad lib)' or something similar.

Fixed: *competitive interactions over food provided ad libitum...*

I. 23-24 "than many" should be "than in many"

Fixed

I. 89 – I recommend the authors provide some interpretation of these values so it is clear what point the authors are trying to make. Is this higher/lower/comparable to average relatedness in groups of the primate species discussed in prior paragraphs?

Fixed. Text now reads:

Kinship within roosts is on average low (0.07-0.08) but highly variable (Wilkinson 1985b; Huguin et al. 2018; Ripperger et al. 2019) similar to other mammals with high relationship complexity (Lukas and Clutton-Brock 2018). For example, the same average kinship is seen in female yellow baboons, Eastern gorillas, and Asian elephants (Lukas and Clutton-Brock 2018).

I. 92 – Is “recruit” in this sentence used to refer to immigration, reproduction, or immigration + reproduction?

Fixed: *an adult female might immigrate into a group about every two years*

I. 95-97 – Is it known whether dominance in male vampire bats is associated with the potential predictors of dominance in females tested in this study (age, body size)? If so, it would be useful to mention here what is known about the basis of dominance in males.

Unfortunately, this is unknown.

I. 127 “to that of” should be “to those of”

Fixed

I.206 – 207: Perhaps the authors can elaborate on what they mean here. How does measuring steepness and shape of the hierarchy shed light on uncertainty regarding the estimated hierarchy?

Fixed. Now reads:

In addition to allowing comparisons between datasets, plotting confidence intervals on the winning probability of dominant individuals provides insight into the certainty that an estimated hierarchy reflects a real underlying dominance structure.

I. 212 – “Macdonald” should be “McDonald”

Fixed

I.222 – The authors should provide some more detail on how these methods were implemented. There are variants to these methods and relevant parameters (Dij vs Pij for David’s Score, K parameter for Elo, starting scores, etc.) that should be reported here or in the supplement.

To provide full details, we are publishing our annotated R scripts for the analysis on Figshare. We added the following link:

***All data and R code can be found on Figshare:
<https://doi.org/10.6084/m9.figshare.14043794.v1>***

l.224-226 – the authors introduce Glicko rating because of its ability to assess the confidence of rank estimates, but then they never use this confidence estimate in the remainder of the paper. Could these estimates of uncertainty add some additional insight to your results?

We deleted this text.

Although we did use both Glicko and randomized Elo-rating to assess confidence in the ranks, both analyses were consistent with the observation that ranks could not be assigned with certainty and the same result is shown more clearly by the hierarchy shape plot. Interpreting a confidence interval on a rank depends on understanding the ranking method, whereas every reader can interpret a confidence interval on the probability of winning. The pros and cons of different ranking methods have been reviewed extensively and is beyond the scope of this paper.

l.256-257 – The authors should provide more detail about this custom permutation. At first I thought the subsequent sentence was doing this, but after re-reading these sentences I understood that they actually relate to different questions (1st sentence: do higher-ranked females feed earlier; 2nd sentence: are conflict rates predicted by kinship or other networks). So additionally, it would be good to make it clearer that the second sentence is describing a subsequent test, not explaining the test introduced in the previous sentence.

We moved this text into a different paragraph and it now reads:

To assess whether social rank influenced the time when bats were observed at feeders (e.g. if higher-ranked females fed earlier), we used a permutation test in which we compared the observed regression coefficient for the effect of an actor's rank on event time with the expected coefficient values when each actor was re-assigned a random rank throughout the entire dataset.

l. 265-266 – this sentence is confusing because the authors' justification for this analysis (explain asymmetries) is immediately contradicted by results presented in the following clause (very little asymmetry). My suggestion is to remove the result, which confuses the sentence and shouldn't really be in the Methods section anyway.

Fixed. We removed this text.

l. 275-278 – This explanation of helping up the hierarchy is confusing. Could the authors reword this or break it into multiple sentences?

Thank you for pointing this out. Revised text now reads:

For each female, we assessed the total proportion of help that was directed up the hierarchy and tested if this was different than 50%. We also measured the mean difference in the interaction rate for partners that were higher rank vs lower rank and tested if this was different than zero. In both cases, we tested this by calculating a 95% confidence interval around the mean using bootstrapping (5000 iterations, percentile method in the boot package (Canty and Ripley 2015; Puth et al. 2015)).

l. Fig 2 – I was curious about the criteria for the selection of these 15 species, so I had a look at some of these studies. In my investigation, I found that the Jenks et al (1995) matrix contains males, so I believe it should not be included in this analysis.

Thank you! We removed this species and re-created this Figure.

l. 324 – “than the” should be “that the”

Fixed

l.336-337 – I’d add “except where noted” because the authors do note somewhere a difference in results based on ranking methods.

Fixed.

l. 358 – “and or” should be “or”

Fixed

Figure 4. Why does the y-axis range between -100 and 100? The authors use normalized David’s score, which should range from 0 to N, where N is the number of individuals in the group. Or are these un-normalized scores?

Fixed. Plot now shows normalized scores.

Figure 5. The y-axis and figure caption suggest that the left panel indicates that some individuals spend 100% of their time sharing food up the hierarchy. Is it not instead that some individuals direct 100% of their food sharing time up the hierarchy, while others direct none of their food sharing time up the hierarchy?

Fixed. Y-axis now just reads “proportion” and figure caption says “Left panel shows the proportion of grooming or food-sharing that each female (blue circle) directed to higher-ranking females.”

l. 412 – “this making” should be “this by making”

Fixed

l. 440-442 – A related point that I would like to see discussed (I think it could fit here or in prior paragraphs) is that vampire bats share the exact food that they are competing over. It would make sense to me that a dominant would step aside to allow a subordinate individual to feed if the alternative was that that dominant had to regurgitate blood to the same subordinate a few hours later. I guess what I’m saying is that if the “currency” of cooperation and competition is identical, then there is less reason for dominants to use their preferred social status to monopolize the currency. A way to resolve this would be to test dominance in contexts other than competition over food (for instance, competition over heat source as suggested on line 414).

Revised text now states:

The fact that vampire bats regurgitate food to feed their close female associates suggests a lack of competition over food with preferred partners (Wilkinson 1984; Carter and Wilkinson 2013a). That is, if an individual would give food to another, there is no reason to use rank to monopolize that food.

Discussion – One point I found missing from the Discussion was what implications this attenuated hierarchy has for the biology of vampire bats or dominance in general. If these results are representative of vampire bat behavior, what does this mean about the nature of competition in this species? The discussion on lines 432-452 kind of gets at this point, but it is mostly framed in the context of why the results of the study might be expected. Maybe this paragraph can be split to have a dedicated paragraph interpreting the significance of the results of the paper.

We divided this paragraph, and added the following text:

Fitness interdependence involves a correlation between the fitness of different individuals, but it is hard to distinguish cause from effect. Cooperative food-sharing relationships could lead to reduced competition over food among familiar bats, or alternatively, a lack of competition over food between frequent roostmates could facilitate the evolution of cooperative relationships.

Reviewer: 2

Comments to the Author(s)

I enjoyed reading the manuscript. I have no comments. The doubts that I was writing down are evacuated later in the discussion in the part that the authors warn about the considerations of the study.

We are glad you enjoyed the manuscript!

My impression with vampires is that spatial memory to find food must be a valuable resource within a group. This spatial memory could be better with experience (age?) And this could possibly lead to hierarchies, however it is difficult to prove this in captive designs.

We thank both reviewers for their comments.

Appendix B

Author responses to comments in bold.

--Gerald Carter, Laurent Brent, and Rachel Crisp

Note to editor: there was a problem either with the submission software or in how I submitted the manuscript, and the paper was not re-submitted although I did get a confirmation that it was submitted on May 7. So the revised manuscript (with minor revisions) was sitting in limbo for awhile. After emailing the journal to reallow submission, I am now resubmitting on June 4, 2021.

Kind regards,

Anita Kristiansen
Editorial Coordinator

on behalf of Dr Oliver Schülke (Associate Editor) and Kevin Padian (Subject Editor)
openscience@royalsociety.org

Associate Editor Comments to Author (Dr Oliver Schülke):

Comments to the Author:

Dear Gerry, Lauren and Dr. Crisp,

thanks for making the effort to revise the paper for RSOS. I am impressed by your turn-around times!

I still have some comments regarding the framing of the study. If you are able to resolve these issues I will be happy to accept it.

We thank you for taking additional time to help improve our paper. We agree with the points you make. We are certainly happy to spend less time on discussing the possible application or misapplications of models and theory developed for primates, and to leave those interpretations for the reader and for future work. Our aim here was simply to test the extent to which female vampire bats have a clear dominance hierarchy and whether that correlates with cooperative behavior. A main reason we sent this paper to RSOS was to avoid any pressure to oversell our results and to be free to discuss all limitations at length. As you say, we hope to better explore the role of competition for resources and services but this study is only the first step, and we leave this topic for future studies.

Line 45: Instead of just saying "For example" here I suggest to introduce primates as an order where this has been extensively studied. Then it is clear why you stick with primates for a bit.

Revised text (line 40):

In many primates, where social ranks and social structures of natural populations are often well documented, reproductive success is influenced by a female's network of long-term individualized social relationships, each involving a mix of cooperation and conflict (5–10). Long-term field studies of rhesus macaques (*Macaca mulatta*) and baboons (*Papio* sp) show that females form highly stable and strongly linear dominance hierarchies with clear

hierarchical relationships between individuals and between matriline (11,12), and that female social rank predicts female fitness (8,13–15).

Line 66ff (line nrs. refer to the version with racked changes): Here you simply state how grooming patterns change with steepness of the dominance hierarchy but you do not say why this would be the case. You say with low hierarchies you benefit more from grooming who grooms you most. What is the benefit of that and what is the benefit of grooming up the hierarchy in a steeper system? You talk about rank regulating access to resources but there is one logical step missing which is tolerance or passive food sharing being interchanged for grooming. If you are to discuss the lack of direct competition over food as the reason for the lack of a clearly developed dominance hierarchy in your bats, you need to explain here what competition has to do with all of this. But since you do not want to go there (see below), you better omit any mentioning of benefits or competition here – just describe the patterns.

In lines 66-74, we removed discussion of benefits and competition and we now simply describe observed patterns.

Original text:

For example, the relationship between social rank and social grooming might depend on the “steepness” of the dominance hierarchy (22,23). In steep hierarchies where social rank determines access to resources like food, low-ranked individuals might benefit from directing grooming up the hierarchy (16,24,25), but in shallow dominance hierarchies where resources cannot be monopolised, they might benefit more from grooming those who groom them more (23,26–29). It remains unclear whether these primate-based models can be applied to non-primates that also form complex individualized cooperative relationships (but see (30)). The goal of this study is to assess (1) if female vampire bats have clear dominance ranks and (2) if rank influences social grooming or food sharing.

Revised text (line 50):

For example, in groups with steep hierarchies, low-ranked individuals often direct grooming up the hierarchy, whereas in groups with shallow dominance hierarchies, grooming is often more symmetrical (16–29). It remains unclear whether such patterns apply to non-primates that also form complex individualized cooperative relationships (but see (30)).

Line 71: I understand now what you are trying to do, but I still have problems with that. I think that you still do not say clearly enough what you want to do. You still say: It remains unclear whether these primate-based models can be applied to non-primates. If you are to answer this, you need to say what “these models” are. If you want to understand whether they are applicable, you do not need to test their predictions, you need to assess their assumptions and inner workings and decide whether vampire bat biology fits those (see also line 468 in discussion). If you decide they do, the models can be applied and the next step is to test their predictions. BUT I think what you are really doing is testing whether similar patterns of grooming distribution according to dominance rank can be found in bats. Which would be fine, but not if you want to discuss them in relation to competition. If you want to go there, you have to explain the role of competition in “these models”.

See above revisions. To better clarify the focus on vampire bats as their own topic, rather than as a comparison of models of primate grooming, we also removed the phrase “interesting comparison with primates” from the next paragraph.

Original:

The social lives of common vampire bats make an interesting comparison with primates, because they seem to share many convergent life history and social traits despite their lineages diverging about 60 - 70 million years ago (32).

Revised text (line 58):

Common vampire bats seem to share many convergent life history and social traits with primates despite their lineages diverging about 60 - 70 million years ago (32).

Line 73: If these are indeed your goals, you do not need any theory or models for that. You can simply say that hierarchy characteristics have been extensively studied in other species, that dominance rank can have profound effects on the exchange of grooming and other goods and services which may be modulated by how steep the hierarchy is. This way you stay on the phenomenological level, you can maintain your aims and you do not have any troubles with how you present the models and how they relate to this work (being assessed for applicability, being tested, or else). This would also go well with the tests you on lines 158ff. You could also largely maintain your discussion. Just describe the patterns found and how they relate to what is known from other species and then say that competition for resources and services is a key factor in theoretical models and that you have not studied this here but that you want to explore in the following where this could go.

We have revised sections of the discussion along the lines suggested by the editor. We removed references to models and theory from the primate literature (underlined), and made minor clarifications.

Original:

Contrary to patterns in some female primates (23,24), we found no evidence that females directed grooming or food sharing to higher-ranking females, or that rates of grooming, food sharing, or their symmetry, were biased to closer ranked females. As predicted where food resources cannot be monopolized or dominance hierarchies are subtle (23), we found no clear support for the hypothesis that grooming or food sharing is exchanged for tolerance from higher-ranked females (e.g. 76). This finding is consistent with other evidence that grooming and food sharing promotes reciprocal investments, rather than tolerance or access to other resources (31,39,40,43). However, a lack of a clear relationship between cooperative relationships (based on food-sharing and grooming) and rank (based on interactions at feeders) is not clear evidence for no relationship between these behaviours or that these primate-based models cannot be applied to vampire bats. The findings of our study must be interpreted with several important limitations in mind.

Revised (line 349):

Contrary to patterns in some female primates (23,24), we found no evidence that females directed grooming or food sharing to higher-ranking females, or that rates of grooming, food sharing, or their symmetry, were biased to closer ranked females. We also found no clear evidence that grooming or food sharing is exchanged for tolerance from higher-ranked females (e.g. 76), which is consistent with other evidence that grooming and food sharing promotes reciprocal investments, rather than mere tolerance (31,39,40,43). However, a lack of a clear relationship between cooperative relationships (based on food-sharing and grooming) and rank (based on interactions at feeders) is not clear evidence for no relationship between these behaviours. The findings of our study must be interpreted with several important limitations in mind.

I am sorry for creating extra work for you but I really think that a bit more clarity is needed here to guide your readers and make them take away a maximum of info.

Take care,
Ole

We appreciate your additional suggestions for further clarification. With your permission, we would also have added your name to the acknowledgements rather than simply “the editor”.